# Salidroside: A Promising Agent in Bone Metabolism Modulation

**DOI:** 10.3390/nu16152387

**Published:** 2024-07-23

**Authors:** Piotr Wojdasiewicz, Stanisław Brodacki, Ewa Cieślicka, Paweł Turczyn, Łukasz A. Poniatowski, Weronika Ławniczak, Mieszko Olczak, Elżbieta U. Stolarczyk, Edyta Wróbel, Agnieszka Mikulska, Anna Lach-Gruba, Beata Żuk, Katarzyna Romanowska-Próchnicka, Dariusz Szukiewicz

**Affiliations:** 1Department of Biophysics, Physiology and Pathophysiology, Faculty of Health Sciences, Medical University of Warsaw, Chałubińskiego 5, 02-004 Warsaw, Poland; piotr.wojdasiewicz@wum.edu.pl (P.W.); stanislaw.brodacki@wum.edu.pl (S.B.); edyta.wrobel@wum.edu.pl (E.W.); agnieszka.mikulska@wum.edu.pl (A.M.); beata.zuk@wum.edu.pl (B.Ż.); katarzyna.romanowska-prochnicka@wum.edu.pl (K.R.-P.); 2Department of Rehabilitation, St. Anna’s Trauma Surgery Hospital, Mazovian Rehabilitation Center—STOCER, Barska 16/20, 02-315 Warsaw, Poland; ew.cieslicka@gmail.com (E.C.); lach.gruba@onet.pl (A.L.-G.); 3Department of Early Arthritis, Eleonora Reicher National Institute of Geriatrics, Rheumatology and Rehabilitation, Spartańska 1, 02-637 Warsaw, Poland; pawel.turczyn@spartanska.pl; 4Department of Neurosurgery, Dietrich-Bonhoeffer-Klinikum, Salvador-Allende-Straße 30, 17036 Neubrandenburg, Germany; lukasz.poniatowski@gmail.com; 5Health Department, Institute of Health Holispace, ul. Św. Wincentego 93/5, 03-291 Warsaw, Poland; weronika.lawniczak@holispace.com; 6Department of Forensic Medicine, Center for Biostructure Research, Medical University of Warsaw, Oczki 1, 02-007 Warsaw, Poland; 7Spectrometric Methods Department, National Medicines Institute, 30/34 Chełmska, 00-725 Warsaw, Poland; e.stolarczyk@nil.gov.pl

**Keywords:** salidroside, osteoporosis, *Rhodiola rosea*, fracture healing, osteoarthritis, adaptogens

## Abstract

*Rhodiola rosea*, a long-lived herbaceous plant from the Crassulaceae group, contains the active compound salidroside, recognized as an adaptogen with significant therapeutic potential for bone metabolism. Salidroside promotes osteoblast proliferation and differentiation by activating critical signaling pathways, including bone morphogenetic protein-2 and adenosine monophosphate-activated protein kinase, essential for bone formation and growth. It enhances osteogenic activity by increasing alkaline phosphatase activity and mineralization markers, while upregulating key regulatory proteins including runt-related transcription factor 2 and osterix. Additionally, salidroside facilitates angiogenesis via the hypoxia-inducible factor 1-alpha and vascular endothelial growth factor pathway, crucial for coupling bone development with vascular support. Its antioxidant properties offer protection against bone loss by reducing oxidative stress and promoting osteogenic differentiation through the nuclear factor erythroid 2-related factor 2 pathway. Salidroside has the capability to counteract the negative effects of glucocorticoids on bone cells and prevents steroid-induced osteonecrosis. Additionally, it exhibits multifaceted anti-inflammatory actions, notably through the inhibition of tumor necrosis factor-alpha and interleukin-6 expression, while enhancing the expression of interleukin-10. This publication presents a comprehensive review of the literature on the impact of salidroside on various aspects of bone tissue metabolism, emphasizing its potential role in the prevention and treatment of osteoporosis and other diseases affecting bone physiology.

## 1. Introduction

*Rhodiola rosea* (*R. rosea*), a member of the *Crassulaceae* family, is named after the Greek words “rodia” or “rodion”, which refer to its pink color when dried. It goes by various names such as “golden root” due to its shiny appearance after purification and “arctic root” because of its habitat in cold regions. This small perennial herb thrives in harsh environmental conditions and low temperatures. It can be found in polar regions across North America, Europe, and Asia, characterized by compact stem leaves up to 40 cm tall and roots with thick rhizomes that release a distinct rosy fragrance when crushed. These rhizomes contain extracts responsible for the plant’s medicinal properties [1]. *R. rosea* is categorized as an adaptogen, meaning it contains active components that help the human body adjust to environmental stress factors. Its properties were utilized for centuries in cases such as chronic stress, fatigue, insomnia, and concentration problems [2,3]. *R. rosea* has been the subject of numerous studies aimed at understanding its mechanisms of action and establishing its potential role in medicine [4,5]. It was noted early on that the medicinal properties of this substance are largely attributed to salidroside (SAL), a compound with various beneficial effects [6]. SAL belongs to the group of Phenylethanoid glycosides and is distinguished by its high biological availability among compounds in this group due to its relatively simple structure [7]. Additional compounds such as rosarin, rosin, and rosavin have been uncovered in subsequent studies and contribute to the unique properties of *R. rosea* [8]. However, SAL continues to garner the most attention, with increasing research on its effects annually [9]. Extensive analysis of this compound involves studying its regenerative and anti-inflammatory effects [10]. Other studies have demonstrated SAL’s beneficial effect on pulmonary fibrosis in animals, protection of cardiac muscle from exhaustive exercise-induced injury, inhibition of myocardial remodeling following a heart attack, and neuroprotective effects in Parkinson’s and Alzheimer’s diseases, while also providing protection against ischemic stroke [11]. SAL exhibits anti-inflammatory actions in liver fibrosis progression and mitigates nonalcoholic fatty liver disease caused by a lipid-rich diet, while significantly influencing metabolic processes in bone tissue (demonstrating multifaceted osteoprotective effects) [12]. This last property could be most valuable, considering the steadily rising number of patients with diminished bone mineral density (BMD). The number of bone fractures due to osteoporosis (OP) in European Union countries reached 3.5 million in 2010 and is predicted to rise to 4.5 million cases by 2025, resulting in heightened research efforts focused on the prevention and treatment of this condition [13,14]. The escalating numbers not only reduce the quality of life for each patient but also burden the healthcare system with enormous costs. In 2019, Great Britain, Switzerland, and EU countries combined allocated more than EUR 56 billion for OP and osteoporotic fractures, sparking increased interest in substances that could offer new and more effective methods for treating bone tissue diseases [15]. This study aims to systematically summarize the current knowledge on SAL as a promising molecule for preventing and treating OP and other bone diseases. The authors will focus on presenting the critical mechanisms by which SAL exerts influences at cellular and molecular levels, based on research conducted in vitro and in vivo, and will include an exploration of its potential applications in clinical settings [16].

## 2. Biochemical Structure of Salidroside

SAL is a chemical compound classified in the group of phenylpropanoid glycosides with the molecular formula C_14_H_20_O_7_ [17]. Its full chemical name is 2-(4-hydroxyphenyl)ethyl O-β-D-glucopyranoside [18]. The biochemical structure of SAL consists of an aglycone (tyrosol), which features a phenyl skeleton with a hydroxyl group (-OH) at the para position (4) and an ethyl group (-CH_2_CH_2_OH) attached to the phenyl ring [19]. The aglycone is linked to a glucose molecule via a glycosidic bond, forming a β-D-glucopyranoside [20]. Additionally, SAL is water-soluble [5]. The chemical structure of SAL is illustrated in Figure 1.

## 3. Influence of Salidroside on Bone Metabolism

### 3.1. In Vitro Studies

#### 3.1.1. Proliferation and Viability of Osteoblast Precursors

Chen et al. investigated the impact of SAL on the growth of C3H10T1/2 cells (mouse pluripotent mesenchymal stem cell-like fibroblasts) and MC3T3-E1 cells (osteoblast precursors derived from mouse calvariae) [21]. The cells were cultured with varying levels of SAL for 48 h, and their proliferation was assessed using a colorimetric method to evaluate metabolic activity. In C3H10T1/2 cells, SAL caused a slight proliferation increase (up to 12% compared to control) at concentrations ranging from 0.5 µM to 50 µM, while in MC3T3-E1 cells, the highest observed growth increase was 138% [21]. Notably, SAL also markedly enhanced the proliferation of other cells similar to osteoblasts derived from various rodent species, such as rat bone marrow-derived mesenchymal stem cells (rBMSCs) [22,23,24].

In the context of overall bone tissue activity, it is important to consider not only the positive effects of SAL on osteoblast proliferation but also its potential impact on their viability. This question was indirectly addressed by Xie et al., who also used MC3T3-E1 cells in their experiments [25]. In the cited study, it was found that even high concentrations of SAL (100 μM) added to the cell cultures did not reduce the viability of osteoblast precursors compared to the control group cells receiving only an osteogenic induction medium (OIM) containing 50 μg/mL ascorbic acid and 10 mM β-glycerophosphate. Despite the lack of noticeable proliferation in the examined cells, the results were deemed promising, indicating that SAL exhibits potentially very low cytotoxicity and, therefore, potential safety for future therapies.

In the subsequent part of the study, the MC3T3-E1 colonies also received dexamethasone, and it was found that SAL was unable to counteract the negative effects of dexamethasone on osteoblast proliferation over the three-day study period. On the other hand, the colonies that received both dexamethasone and SAL exhibited a higher number of cells in the active phase of cell division compared to the group treated with dexamethasone alone, although these results were not statistically significant. These findings may suggest a potential property of SAL to counteract the effects of steroids on osteoblast proliferation, but it is likely that the observation period of just three days was insufficient to observe significant changes in colony numbers, thus requiring further studies.

#### 3.1.2. Activation and Expression of Bone Morphogenetic Proteins 

SAL was evaluated as a potential activator of bone morphogenetic protein 2 (BMP-2), a crucial factor in osteoblast growth and maturation. Using a luciferase reporter gene assay, various concentrations of SAL were added to MC3T3-E1 cells for 48 h, and luciferase levels were measured. A clear enhancement in reporter gene expression was detected for SAL concentrations ranging from 0.25 µM to 10 µM [21]. In MC3T3-E1 cultures, 1 µM of SAL administered for 6 and 12 days markedly increased BMP-2 mRNA levels as identified by RT-PCR. Additionally, SAL treatment in MC3T3-E1 and C3H10T1/2 cultures triggered rapid phosphorylation of Smad1/5/8 and enhanced extracellular signal-regulated kinase 1/2 (ERK1/2) pathway stimulation, as confirmed by other in vitro studies [22]. Co-treatment with noggin, a BMP antagonist, or dorsomorphin, an inhibitor of the type I BMP receptor, significantly reduced SAL-induced ALP levels and Smad1/5/8 phosphorylation (markers of osteoblasts maturation) [21]. To further determine whether SAL influences BMP mRNA levels during osteoblast differentiation, 5 µM SAL was administered to C3H10T1/2 cultures for 18 days, resulting in elevated expression of BMP-2, BMP-6, and BMP-7 [21]. It is worth noting that BMP-6 and BMP-7 proteins, similar to BMP-2, play a crucial role in regulating bone tissue metabolism. It has been proven that BMP-7 supports the fracture healing process by accelerating bone tissue regeneration through the stimulation of osteoblast differentiation. On the other hand, BMP-6 influences bone homeostasis by modulating the activity of osteoclasts (including inhibiting their activity), which is key to inhibiting the resorption process [26,27].

#### 3.1.3. Alkaline Phosphatase Activity and Mineralization

Alkaline phosphatase (ALP) is an indicator of osteoblast maturation. C3H10T1/2 cells exposed to SAL (0.5–10 µM) for 12 and 18 days exhibited a substantial rise in ALP activity, with the highest elevation seen with 5 µM of SAL at 18 days. Alizarin Red S staining demonstrated enhanced osteoblast mineralization after 21 days of SAL exposure [21]. Additionally, mRNA expression of osteoblast indicators ALP and osteopontin, along with bone-forming transcription factors like runt-related transcription factor 2 (Runx2) and osterix (OSX), were markedly increased in C3H10T1/2 cells treated with 5 µM of SAL for 18 days, as measured by reverse transcription–polymerase chain reaction (RT-PCR). Comparable effects were noted in MC3T3-E1 cells exposed to SAL (0.5–5 µM) for 12 days [21].

#### 3.1.4. Adenosine Monophosphate-Activated Protein Kinase Activation

Fu et al. investigated the function of adenosine monophosphate-activated protein kinase (AMPK) in SAL-induced osteoblast growth and differentiation utilizing MC3T3-E1 cultures [28]. The samples were divided into six sets and maintained in osteogenic medium for 14 days. Three sets were exposed to varying levels of SAL (1 µM, 5 µM, 10 µM) for 48 h. Another set was administered 1 mM of 5-aminoimidazole-4-carboxamide ribonucleotide (AICAR), an AMPK stimulator, for 48 h. A fifth set received 10 µM of SAL and 40 µM of Compound C, an AMPK inhibitor. The control set was maintained in osteogenic medium without additional treatments. Culture growth was evaluated by recording the optical density (OD). The results indicated that SAL promoted MC3T3-E1 sample expansion in a dose- and time-dependent fashion. After 24 h, cultures exposed to 5 µM and 10 µM of SAL exhibited markedly higher OD values relative to the control and 1 µM SAL sets. After 36 h, all SAL-exposed sets demonstrated notably enhanced expansion. The AICAR set also displayed a notable rise in OD values at 24 and 36 h. Co-treatment with Compound C and 10 µM of SAL notably diminished the pro-growth effect of SAL relative to 10 µM of SAL alone [28]. ALP activity, an indicator of osteoblast differentiation, was markedly elevated in all SAL-exposed sets relative to the control, with the greatest elevation seen in the 10 µM SAL set. The smallest rise was noted in the set administered SAL and Compound C. Alizarin Red assay results demonstrated that SAL enhanced the formation of calcium nodules, with the largest number in the AICAR set and the fewest in the SAL and Compound C set. Western blot analysis revealed that the levels of bone-related proteins collagen type I alpha 1 (COL1A1), osteocalcin (OCN), and RUNX2 were markedly upregulated in the AICAR and SAL-exposed sets, particularly in the 10 µM SAL set. However, the SAL and Compound C set exhibited a reduced effect. Furthermore, phosphorylated AMPK (*p*-AMPK) levels were markedly elevated in the AICAR- and SAL-exposed sets relative to the control, whereas the SAL and Compound C set presented a decreased *p*-AMPK/AMPK ratio [28].

#### 3.1.5. Cellular and Molecular Impact on Endothelial Cells

Vascular endothelial cells are essential in bone regeneration, as they influence the bone microenvironment. Endothelial progenitor cells (EPCs) have shown promise in treating vascular diseases related to bone tissue by restoring endothelial function and aiding vascular repair [29]. Vascular endothelial growth factor (VEGF) plays a crucial role in angiogenesis and the survival of endothelial cells, which are vital for bone tissue regeneration and repair [30]. VEGF enhances the vascularization of bone by stimulating the formation of new blood vessels within bone tissue, ensuring the supply of necessary nutrients and oxygen to the healing area [31]. This is particularly important for the treatment and prevention of OP, as improved vascularization in bone leads to higher BMD and strength, thereby decreasing the risk of fractures [32]. Additionally, lifestyle interventions, including diet and exercise, support endothelial function and improve the effectiveness of OP treatments by promoting bone regeneration and density [33]. These mechanisms highlight the critical role of the vascular endothelium in bone regeneration and the treatment of bone disorders. Research indicates that SAL has anabolic effects on endothelial cells, making it a promising agent for enhancing bone tissue metabolism in various diseases. The detailed cellular mechanisms of this action in in vitro studies are discussed below.

##### Improvement of Proliferation and Viability of Endothelial Cells

To evaluate the impact of SAL on the survival and growth of endothelial cells, Guo et al. conducted a series of in vitro experiments [34]. The researchers used the human endothelial cell line EA.hy926 and human umbilical vein endothelial cells (HUVECs) exposed to conditioned medium (CM) obtained from osteoblast cultures (MG-63 cells) that had previously received standard medium containing 1% charcoal-stripped FBS for the first 24 h and various concentrations of SAL (0, 10 nM, and 100 nM) for the next 24 h [33]. The results revealed that CM from MG-63 cultures treated with 100 nM of SAL significantly and statistically increased endothelial cell growth and viability by up to 126% compared to endothelial cells that received CM from MG-63 cells without SAL (after a 48 h incubation). Notably, there were no significant differences between the effects of CM containing 10 nM and 100 nM SAL on both HUVEC and EA.hy926 cells, suggesting a plateau in the dose–response relationship at these concentrations. This implies that even at lower concentrations, SAL can indirectly and effectively promote endothelial cell growth and survival, making it a potent agent for enhancing angiogenic processes [34].

The lack of a noticeably greater impact of SAL at concentrations higher than 10 nM is likely related to the biological maximum capacity of the studied cells to respond to stimulation at this level. Considering the multifaceted influence of SAL on the activation of anabolic processes in bone tissue, a dose above 10 nM may not generate a greater biological effect beyond the genetic and physical capabilities of the cell. This observation might also indicate that SAL does not overstress cellular metabolism, which could otherwise lead to apoptosis or, in a worse case, contribute to proliferative changes. This is also potentially good news in the context of determining the maximum dosage of SAL for human use in the future, as it suggests that SAL may exhibit optimal efficacy at relatively low doses.

##### Stimulation of Migration and Capillary Tube Formation

Cell migration is a vital process in angiogenesis where endothelial cells move to form new blood vessels. Guo et al. assessed the impact of SAL on endothelial cell movement using conditioned medium (CM) with varying amounts of SAL [34]. The results indicated that CM alone could promote the movement of EA.hy926 and HUVECs. However, when SAL was added to the CM at levels of 10 nM and 100 nM, there was a notable boost in cell movement. Specifically, the movement of EA.hy926 cells increased by approximately 50%, while HUVECs showed an increase of about 30% compared to cells exposed to CM alone [34].

To further understand the role of VEGF in this process, anti-VEGF antibodies were introduced to the culture. The addition of anti-VEGF antibodies greatly reversed the movement induced by CM with SAL, while the use of isotype control antibodies had no such effect. This reversal suggests that the angiogenic activity of SAL is heavily mediated by VEGF levels [34].

The study also investigated the effect of SAL on capillary tube formation in vitro using HUVECs as a model. The ability of endothelial cells to form capillary structures is a hallmark of angiogenesis. The results demonstrated that both 10 nM and 100 nM of SAL in the CM greatly enhanced the formation of capillary tubes compared to CM alone. The length of the capillary tubes in the presence of SAL was 46% greater than in the CM-only group. Similar to the movement assays, the application of anti-VEGF antibodies notably diminished the tube formation induced by SAL, confirming the main role of VEGF in this process [34].

##### Activation of the HIF-1α/VEGF Signaling Pathway

To delve deeper into the molecular mechanisms of angiogenesis in bone tissue, Guo et al. analyzed the levels of HIF-1α and VEGF in endothelial cells treated with SAL [34]. Western blot and RT-PCR analyses were employed to estimate the mRNA and protein levels of these factors. The data revealed that SAL treatment significantly elevated the levels of HIF-1α and VEGF mRNA and proteins in EA.hy926 cells, with the most notable effects observed at the 10 nM concentration. This suggests that SAL can effectively enhance the expression of these key angiogenic factors at relatively low concentrations [34].

Further experiments were conducted to test the specificity of this response using 3-(5′-hydroxymethyl-2′-furyl)-1-benzylindazole (YC-1), a known inhibitor of HIF-1α. EA.hy926 cells were treated with YC-1 before being exposed to SAL. The findings showed that YC-1 substantially reduced the SAL-induced levels of HIF-1α and VEGF. However, SAL was able to partially counteract the inhibitory effects of YC-1, indicating that SAL mitigates the suppression of HIF-1α and VEGF expression [24]. Moreover, luciferase reporter assays were performed to assess the transcriptional activity of HIF-1α in cells exposed to different concentrations of SAL. The assays showed that SAL at 10 nM substantially increased HIF-1α transcriptional activity under hypoxic conditions, further supporting the role of SAL in promoting angiogenesis through the initiation of the HIF-1α/VEGF signaling pathway (Figure 2) [34].

Additionally, Guo et al. explored the cellular and molecular effects of SAL on bone metabolism and the HIF-1α pathway not only in endothelium but also in osteoblasts (cells isolated from neonatal BALB/c mice), potentially influencing fracture healing [35]. In vitro studies demonstrated that SAL significantly promoted cell growth by altering the cell cycle of osteoblastic cells and stimulated bone formation by enhancing their differentiation and mineralization. SAL induced the production of Runx2 and OSX and activated the HIF-1α/VEGF signaling pathway, promoting both angiogenesis and osteogenesis through a non-autonomous (indirect) mechanism. The mitogen-activated protein kinase (MAPK)/ERK route and the phosphoinositide 3-kinase (PI3K)/protein kinase B (Akt) signaling route were found to be essential for SAL-mediated osteoblast growth and HIF-1α levels in MG-63 cells, indicating that SAL exerts its effects on osteoblastic cells by engaging these pathways [35].

#### 3.1.6. Effects of Salidroside on Glucocorticoid-Induced Osteoporosis

Xie et al. examined the protective effects of SAL on glucocorticoid-induced OP, specifically focusing on dexamethasone-induced bone loss [25]. Dexamethasone, a synthetic glucocorticoid, is known for its anti-inflammatory and immunosuppressive properties, but it also inhibits osteoblast activity, reduces osteoprotegerin (OPG) production, decreases collagen synthesis, and impairs calcium absorption. These effects make glucocorticoid-induced OP the third most common type of OP after postmenopausal and senile OP [25]. The researchers utilized the MC3T3-E1 cell line, consisting of pre-osteoblasts isolated from the skulls of *Mus musculus*. The cells were then grown in OIM and treated with dexamethasone (10 μM) and SAL (10 and 20 μM) for three weeks. Alizarin Red S staining, which indicates mineralized nodules, showed that dexamethasone significantly inhibited mineralization. However, this inhibitory effect was markedly reduced by the addition of SAL. Furthermore, SAL activated the transforming growth factor-beta (TGF-β)/Smad2/3 cellular route, essential for maintaining tissue homeostasis and regulating cellular responses to tissue injury and repair. SAL also mitigated the dexamethasone-induced suppression of protein expression levels of OSX, a transcription factor crucial for the maturation of osteoblasts, and significantly enhanced ALP activity and mineralization, promoting osteogenic differentiation. The protective effects of SAL against dexamethasone-induced inhibition of osteogenic differentiation can be blocked by LY2109761, an inhibitor of TGF-β type I/II receptors [35]. Xue et al. conducted studies to investigate the effects of SAL on steroid-induced avascular necrosis of the femoral head (SANFH) using the PI3K/Akt signaling pathway [36]. These studies included in vitro and in vivo methods. In the in vitro experiments, primary rat osteoblasts were treated with SAL and dexamethasone to activate osteoblast apoptosis. The results demonstrated that SAL at concentrations of 10 nM and 100 nM significantly protected osteoblasts from dexamethasone-induced cell death and apoptosis. This protective effect was confirmed in this study through terminal deoxynucleotidyl transferase dUTP nick end labeling (TUNEL) staining, which detected fewer osteoblasts in the form of apoptotic cells, and Western blot analyses, which showed decreased levels of apoptotic markers such as Bcl-2 Associated X-protein (BAX), caspase-3 (CASP3), and caspase-9 (CASP9) in cell cultures. These findings suggest that SAL mitigates apoptosis pathways, thereby protecting osteoblasts [36].

#### 3.1.7. Salidroside’s Role in Mitigating Osteoporosis through Antioxidant Activity

Wang et al. aimed to demonstrate the protective effects of SAL against OP by inhibiting oxidative stress and encouraging osteogenesis in a mouse model of ovariectomy (OVX)-induced estrogen deficiency [37]. OP is characterized by low bone mass, deterioration of bone microarchitecture, and increased bone brittleness. Contributing factors include aging, reduced levels of estrogen, vitamin D, and calcium, and elevated levels of reactive oxygen species (ROS). Evidence suggests that the balance between ROS and antioxidants plays a notable role in the development of OP [37]. In vitro, the researchers used a tert-butyl hydroperoxide (t-BHP)-induced oxidative stress model in rat osteoblasts (ROBs) to assess the defensive capabilities of SAL against cell apoptosis, osteogenic differentiation, antioxidant capacity, and nuclear factor erythroid 2-related factor 2 (Nrf2) production. ROBs were extracted from the skulls of 4-week-old rats and grown. The cells were exposed to various levels of t-BHP (50–100 μM) for six hours, and different levels of SAL (0.01–100 μM) were administered for 24 or 48 h. Following pretreatment with SAL, the medium was replaced with t-BHP for six hours. Cell viability was evaluated, and flow cytometry and Western blotting were used to analyze apoptosis and protein production. The results showed that SAL pretreatment significantly protected ROBs from t-BHP-induced oxidative damage and apoptosis. SAL encouraged osteogenic differentiation, enhanced antioxidant capacity, and increased Nrf2 production while reducing Kelch-like ECH-associated protein 1 (Keap1) production, which is an inhibitor of Nrf2. These findings indicate that SAL exerts its defensive effects by activating the Nrf2 pathway and mitigating oxidative stress [37].

In another study, Zhang et al. assessed SAL’s potential to protect against oxidative stress-induced bone loss by using hydrogen peroxide (H_2_O_2_)-induced oxidative stress in MC3T3-E1 cultures [38]. H_2_O_2_ significantly reduced cell viability, with a 300 µM concentration causing approximately 50% viability reduction after 24 h. Pretreatment with SAL (0.1–10 µM) for 24 h significantly improved cell survival against subsequent H_2_O_2_ exposure. SAL also protected osteoblast function, as evidenced by increased ALP performance and enhanced production of differentiation markers ALP, COL1A1, and OCN, which has also been observed in other in vitro and in vivo studies [22,23,24,28].

### 3.2. In Vivo Studies

#### 3.2.1. Protective Effects against Oxidative Stress

In further studies, this time involving an in vivo method, Zhang et al. [38] aimed to determine the effects of SAL administered intraperitoneally for 15 weeks on oxidative stress markers in the blood of OVX mice. The mice received SAL at doses of 5 mg/kg and 20 mg/kg body weight. Compared to the control group, which did not receive SAL, the SAL-treated mice exhibited elevated glutathione (GSH) levels and reduced malondialdehyde (MDA) levels in the blood, especially in the mice receiving the higher concentration of SAL, which is consistent with the findings of many other researchers on this topic [21,24,37]. This therapy also resulted in improved bone quality in the treated mice, as observed in post-mortem examinations. Micro-computed tomography (micro-CT) imaging of the femoral distal metaphysis and lumbar vertebrae from treated mice showed that SAL (20 mg/kg) significantly preserved bone microstructure parameters, including bone volume/total volume (BV/TV%), trabecular number (Tb.N), trabecular separation (Tb.Sp), trabecular thickness (Tb.Th), connectivity density (Conn.D), structure model index (SMI), and BMD. In addition to imaging studies, histological analyses confirmed that SAL treatment mitigated trabecular thinning and enhanced calcium deposition [38]. It is worth noting that the observed improvement in bone quality in OP models in response to SAL in OVX rodents also corresponds with the findings of other researchers [37], as described in Section 3.2.5.

#### 3.2.2. Effects on Knee Osteoarthritis in Mice

Further in vivo studies were conducted on a knee osteoarthritis (KOA) model in mice [28]. Fifty 6-week-old male mice were allocated into five groups: normal, KOA, SAL, AICAR, and SAL with Compound C. KOA was triggered by intra-articular administration of monoiodoacetate. After three weeks, treatments commenced with SAL (50 mg/kg/day) administered intragastrically, AICAR (100 mg/kg/day), or SAL (50 mg/kg/day) with Compound C (20 mg/kg/day) administered intraperitoneally. The normal and KOA groups did not receive treatment. Mice were weighed weekly, and the arthritis index (AI) was assessed. Body weight increased across all groups, with the highest increase in the normal group and the smallest in the KOA group. AI markedly increased in the KOA model groups relative to the normal group. SAL and AICAR treatments reduced AI after 21 days, while the combination of SAL and Compound C resulted in higher AI values [28]. Enzyme-linked immunosorbent assays (ELISAs) on plasma samples showed elevated levels of the inflammatory indicators tumor necrosis factor-alpha (TNF-α) and interleukin-6 (IL-6) and reduced interleukin-10 (IL-10) in the KOA group relative to the normal group. SAL and AICAR treatments significantly decreased TNF-α and IL-6 levels and elevated IL-10 levels. The combination of SAL and Compound C nullified these anti-inflammatory effects [28]. Western blot analysis of knee joint tissues revealed enhanced production of osteogenic proteins RUNX2, ALP, COL1A1, and OCN in the SAL and AICAR groups relative to the KOA group. The combination of SAL and Compound C reduced this effect. Additionally, *p*-AMPK production was higher in the SAL and AICAR groups relative to the KOA group, but lower in the SAL and Compound C group [28]. Histological analysis showed preserved joint structure with minimal inflammatory cell infiltration in the normal group. The KOA group exhibited bone erosion and cartilage degradation, while SAL and AICAR treatments reduced inflammatory cell presence and preserved joint structure. Tartrate-resistant acid phosphatase (TRAP) staining indicated higher osteoclast activity in the KOA group, which was reduced in the SAL and AICAR groups. The combination of SAL and Compound C increased TRAP production. SAL and AICAR treatments reduced the production of Nuclear Factor of Activated T Cells, Cytoplasmic 1 (NFATc1), which is connected with higher osteoclast function. However, the combination of SAL and Compound C increased NFATc1 production [28].

#### 3.2.3. Angiogenesis in Mouse Embryonic Metatarsals

In another in vivo study, mouse embryonic metatarsal bones were isolated to evaluate endothelial sprouting, a crucial aspect of angiogenesis [34]. The experimental groups included control, SAL, VEGF, SAL + anti-VEGF, and SAL + isotype control immunoglobulin G (IgG). SAL treatment significantly increased the area of endothelial sprouting compared to the control group. The inclusion of anti-VEGF antibodies markedly inhibited this sprouting, emphasizing the essential role of VEGF in mediating the pro-angiogenic effects of SAL. This experiment provided compelling evidence that SAL promotes angiogenesis through VEGF-dependent pathways in an in vivo context [34].

#### 3.2.4. Osteogenesis and Bone Healing

The in vivo aspect of Xue et al.’s study involved adult male rats to evaluate the protective effects of SAL against SANFH [36]. The rats were allocated into control, model, and SAL treatment groups. The model and SAL groups received lipopolysaccharide (LPS) followed by methylprednisolone (MPS) to induce SANFH, while the SAL group additionally received SAL injections. Histological analysis of the femoral heads showed that the SAL-treated group had significantly fewer empty lacunae compared to the model group, indicating a protective effect of SAL against bone necrosis [36]. For another in vivo study, BALB/c mice were used [34]. An isolated fracture was induced in the proximal third of the tibia of these mice, and they were randomly assigned to four groups. Seven days post-surgery, the groups were administered different treatments: a vehicle (fracture model group, FM), 200 μM of SAL (low-dose SAL group, SAL-L), 800 μM of SAL (high-dose SAL group, SAL-H), and 200 μM of desferrioxamine (DFO group). These treatments were administered to the fracture sites every alternate day. Seventeen days after the procedure, radiographic and histological analyses were conducted to evaluate the fracture healing status. The results indicated that SAL significantly enhanced fracture healing. This enhancement was evidenced by improved bone formation and mineralization, increased angiogenesis within the callus, and accelerated overall fracture healing [34].

#### 3.2.5. Effects on Osteoporosis Model in Rats

Ling Li et al. utilized an OVX-induced OP model in rats to study the defensive effects of SAL on bone deterioration [39]. This model mimics postmenopausal OP in humans. The rats were allocated into five groups: control, OVX, OVX + low-dose SAL (4 mg/kg), OVX + high-dose SAL (20 mg/kg), and OVX + raloxifene (RLX), a specific estrogen receptor modulator known to protect against OP. The results showed that SAL significantly increased bone density and mineral apposition rates (MARs), which represent the rate of new bone formation. Serum levels of ALP, OCN, and prostaglandin E2 (PGE2) were also measured. The OVX group exhibited significantly elevated levels of ALP and OCN, indicative of increased bone turnover. However, treatment with high-dose SAL or RLX significantly reduced these levels, suggesting that SAL mitigates the negative effects of estrogen insufficiency on bone turnover [39]. Micro-CT analysis further revealed that SAL enhanced the microarchitecture of trabecular bone. Parameters such as BMD, trabecular area (Tb.Ar), Tb.Th, Tb.N, and Tb.Sp were all significantly enhanced in the SAL-treated groups relative to the OVX group. The histological examination confirmed these findings, showing that SAL partially reversed the negative impact of OVX on bone microarchitecture [37], which is also supported by other in vivo studies [22,23,24]. Additionally, the effects of SAL on osteoid parameters, including osteoid surface (OS), osteoid width (O.Wi), and osteoid volume (OV), were assessed. The OVX group showed increased osteoid parameters, indicative of impaired bone formation, while SAL treatment significantly reduced these parameters, further supporting its defensive role [39]. Immunohistochemical analysis was used to assess the expression of HIF-1α and VEGF in bone tissue. The results showed increased expression of HIF-1α and VEGF in the SAL-treated groups, suggesting that SAL enhances bone healing and regeneration through the HIF-1α/VEGF route [39]. Another study conducted by Zheng et al. on rats provided detailed insights into the significant impact of SAL on the OPG/receptor activator of nuclear factor kappa-Β ligand (RANKL) ratio [40]. The study used female Sprague Dawley rats that had undergone OVX to induce a postmenopausal OP model and were subsequently rendered diabetic through the administration of streptozotocin (STZ). SAL was administered orally at doses of 20, 40, and 80 mg/kg per day for a period of 12 weeks. The treatment significantly upregulated the expression of OPG while simultaneously downregulating RANKL across all dosage groups, with the most pronounced effects observed at the highest dose of 80 mg/kg. Corresponding to these changes, there was a significant increase in BMD in the SAL-treated groups relative to the control group, alongside notable enhancements in overall bone health. These results underscore the potential of SAL as a therapeutic agent in preventing bone deterioration through the modulation of the OPG/RANKL signaling route, particularly in diabetic conditions exacerbated by estrogen insufficiency [40]. The defensive effects of SAL on OVX-induced bone deterioration were also examined in 3-month-old female rats [38]. Four weeks post-OVX, the rats were allocated into groups receiving vehicle treatment, 4 mg/kg/day SAL, or 20 mg/kg/day SAL for 90 days via intragastric injection. Post-treatment histomorphometry of the left tibia revealed that SAL significantly mitigated trabecular bone deterioration, with higher doses providing better protection. SAL treatment also increased trabecular bone volume percentage (TBV%) and reduced trabecular absorption surface percentage (TRS%), indicative of decreased osteoclast activity [38].

## 4. Discussion

The research on SAL presented in this review paper elucidates its multifaceted role in promoting bone health through osteogenesis and angiogenesis. This comprehensive analysis of both in vitro (Table 1) and in vivo (Table 2) investigations highlights SAL’s potential as a treatment option in the treatment of OP and the enhancement of fracture healing.

SAL has been shown to stimulate the growth and maturation of osteoblast precursors, as indicated by the increased activity of ALP and mineralization markers in research conducted by Chen et al. [21]. This is achieved primarily via stimulating the BMP-2 signaling pathway, which is critical for osteoblast differentiation and bone growth. Furthermore, the upregulation of key osteogenic transcription elements such as Runx2 and OSX under SAL treatment reinforces its function in enhancing osteoblast activity and bone formation. These findings can be particularly useful in the field of orthopedics and bone regenerative medicine. A frequent complication after the implantation of endoprostheses is the loosening of implants, which necessitates costly revision surgeries. The field of orthopedics is continually exploring new molecules to coat implants, particularly for high-risk patients and those with low BMD [41]. SAL has the potential to be utilized in such cases due to its beneficial effects on bone tissue mineralization and its ability to promote osteoblast growth and viability.

Fu et al. showed that SAL activates AMPK, a primary controller of cellular energy balance, to promote osteoblast growth and maturation while inhibiting bone resorption [28]. This activation of AMPK is crucial, as it not only supports osteoblast function but also mitigates the effects of bone resorptive conditions such as osteoarthritis (OA). In this context, SAL could potentially prevent frequent subchondral fractures observed in rheumatoid arthritis (RA) and degenerative joint disease, which worsen the prognosis for chronically affected patients and hasten the decision to proceed with endoprosthesis [42]. However, before implementing such an orthopedic procedure, it is often necessary to heal the fracture, typically treated with immobilization. Unfortunately, this approach deteriorates the future treatment prognosis and increases the overall costs.

SAL’s role in endothelial cell function, as investigated by Guo et al., further underscores its another therapeutic potential. By enhancing the HIF-1α and VEGF signaling pathways, SAL promotes angiogenesis, which is essential for bone healing and regeneration [34]. The coupling of osteogenesis and angiogenesis facilitated by SAL ensures adequate vascular support for newly forming bone tissues, enhancing the overall effectiveness of bone repair processes. These insights are valuable for developing advanced therapies in vascularized tissue engineering and regenerative medicine, particularly for conditions such as peripheral artery disease and diabetic foot ulcers. Moreover, the formation of new blood vessels within bone tissue is clinically crucial in situations such as delayed union or the treatment of nonunion fractures. The costs associated with treating delayed union in the USA range between USD 14 and 15 million annually [43]. These complications most commonly occur in locations such as the proximal end of the fifth metatarsal (known as a Jones fracture), the femoral neck (a classic fracture in osteoporotic patients), and the distal third of the tibia. SAL could be a potential molecule for application in the fracture gap during the fixation of fractures at risk of delayed union, especially in patients with conditions such as cachexia, diabetes, alcohol and nicotine dependence, or open fractures.

Xie et al. explored the protective effects of SAL against glucocorticoid-induced OP [25]. Their findings revealed that SAL mitigates the inhibitory effects of dexamethasone on osteoblast viability and differentiation by activating the TGF-β/Smad2/3 signaling pathway [25]. This pathway is crucial for maintaining tissue homeostasis and promoting the differentiation of pre-osteoblasts into mature osteoblasts, highlighting SAL’s potential in counteracting the adverse effects of glucocorticoids on bone health. Xue et al. provided evidence that SAL protects against SANFH through the PI3K/Akt signaling pathway [36]. SAL’s ability to prevent osteoblast apoptosis and promote cell survival under dexamethasone treatment indicates its potential to mitigate steroid-induced bone damage [36]. The implications of these studies are particularly significant for the fields of endocrinology and pharmacology, where the management of long-term steroid side effects is critical, particularly in chronic inflammatory diseases involving joint pathology. Within this group of conditions, another subset of patients who may derive substantial benefits from the therapeutic effects of SAL includes those with RA. These patients are especially prone to osteoporotic fractures induced by prolonged steroid therapy and disease-modifying antirheumatic drugs (DMARDs) [44]. The deterioration of bone tissue metabolism in these patients may also be attributed to the increased pathological secretory function of fibroblasts, which produce numerous inflammatory cytokines directly into the joint space via the synovial membrane. Recent studies indicate that SAL, by indirectly blocking the activity of such fibroblasts, can improve the quality of bone tissue through an additional mechanism [45].

The antioxidant properties of SAL, as detailed by Wang et al., play a significant role in protecting against OP by inhibiting oxidative stress and promoting osteogenesis through the Nrf2 pathway [37]. The activation of Nrf2 enhances the cells’ antioxidant capacity, reducing oxidative damage and promoting bone health, especially in conditions of estrogen deficiency, as modeled in OVX rats. This aspect of SAL is crucial for geriatric medicine and preventive healthcare, focusing on mitigating age-related BMD loss and conditions such as postmenopausal OP. The justification for supplementing *R. rosea* extract, which contains SAL, in postmenopausal women with OP was highlighted in the study of Gerbarg et al. [46], which demonstrates the potential impact of *R. rosea* on Selective Estrogen Receptor Modulators (SERMs), key targets in hormone replacement therapies. The authors of the publication even demonstrate the superiority of SAL over estradiol, which is commonly used during menopause, in terms of certain health aspects, such as a lower risk of oncogenesis. Additionally, the referenced publication notes that *R. rosea* extract positively affects other menopausal symptoms, including cognitive function, memory, energy levels, and mood [46].

As highlighted in the previously cited in vitro studies, SAL impacts multiple signaling pathways that enhance bone tissue metabolism. These pathways are comprehensively summarized in Figure 3.

In vivo investigations further validate SAL’s efficacy in improving bone density, preserving bone microarchitecture, and enhancing bone formation [22,23,24]. For instance, the research by Li et al. demonstrated that SAL significantly increased bone mass and mineral apposition rates in an OVX-induced OP model, indicating its potential as a treatment option for postmenopausal OP [39]. Additionally, SAL’s ability to enhance angiogenesis and accelerate fracture healing through direct (autonomous) and indirect (non-autonomous) mechanisms was demonstrated in mouse models. This supports its potential application in clinical settings for improving skeletal regeneration and repair [35]. The therapeutic properties of SAL extend beyond surgical applications to include rehabilitation. Current evidence suggests that supplementing with SAL can benefit patients with OP even before fractures occur, serving as a preventative measure and enhancing the efficacy of weight-bearing exercises and resistance exercises performed in water [47]. Another group of patients at risk of OP who may benefit from SAL supplementation includes those with spinal cord injury (SCI) and others prone to prolonged immobilization [48,49]. Due to the absence of regular skeletal loading caused by their underlying conditions, there is a pressing need for new, preferably orally administered molecules that can improve bone mass, either as a complement to or independent of rehabilitation efforts [50]. Given the diverse anabolic and anti-inflammatory effects of SAL, which are not restricted to bone cells, it is anticipated that SCI patients may experience improvements not only in BMD but also in other health challenges typical of their condition. These benefits could include alleviation of depression symptoms, reduction in spinal cord edema, and enhancement of glial cell anabolism.

The therapeutic potential of SAL extends to its anti-inflammatory properties in conditions such as KOA. This is demonstrated by its ability to reduce inflammatory markers like TNF-α and IL-6, while increasing the expression of IL-10, thereby improving joint structure in animal models [28]. As previously mentioned, OA and RA are often associated with the formation of subchondral cysts, which can weaken the structural integrity of bones and increase the risk of fractures. This highlights the broader applicability of SAL in managing inflammatory bone diseases and potentially mitigating fracture risks associated with degenerative joint disease. 

Despite the extensive research on SAL, none of the investigations to date have considered its combined effects with rosavin, another bioactive compound that naturally occurs with SAL in *R. rosea*. Rosavin has been demonstrated to possess osteogenic and anti-osteoporotic properties [51,52]. Investigating the synergistic effects of SAL and rosavin could provide a more thorough understanding of their potential in bone health. This unexplored combination may reveal enhanced or complementary mechanisms of action, potentially leading to more effective therapeutic strategies for OP and other bone-related disorders. Future investigations should aim to elucidate the interactions between these compounds to fully leverage the therapeutic potential of *R. rosea* [39,40].

It is important to highlight that the convenience of oral administration of SAL presents a significant advantage. However, research on its bioavailability in the human gastrointestinal tract is essential. Studies should focus on determining how effectively SAL is absorbed and utilized when taken orally. If oral bioavailability proves inadequate, alternative delivery methods, such as transdermal or intravenous administration, may need to be explored to ensure the proper therapeutic concentration of SAL in bone tissue [53,54]. At this point, it is worth mentioning that in some in vitro studies, the therapeutic concentrations of SAL are promising. These studies indicate that SAL exerts its maximum biological effect at relatively low concentrations (10 nM), without additional benefits at higher levels (100 nM) [34]. Thus, it can be cautiously and optimistically assumed that SAL can be safely administered in the future without approaching toxic levels in the prevention and treatment of bone diseases.

Considering the near future, it seems prudent to commence studies on SAL using human cell cultures, including both osteoblasts and osteoclasts, as soon as possible. Confronting the highly promising results obtained from animal models is essential to determine whether further clinical trials of SAL as a potential therapeutic agent for bone diseases are warranted. However, it should be noted that additional in vitro and in vivo studies on animal models should also be conducted to uncover new molecular mechanisms through which SAL may enhance bone tissue metabolism. From the authors’ perspective, this approach to further research on the effects of SAL on bone metabolism would be the most effective.

## 5. Conclusions

SAL presents a promising therapeutic avenue for enhancing bone health and treating various bone-related disorders. Its ability to promote osteogenesis and angiogenesis, coupled with its protective effects against oxidative stress and inflammation, positions it as a valuable candidate for future clinical applications in bone regeneration and OP treatment.

## Figures and Tables

**Figure 1 nutrients-16-02387-f001:**
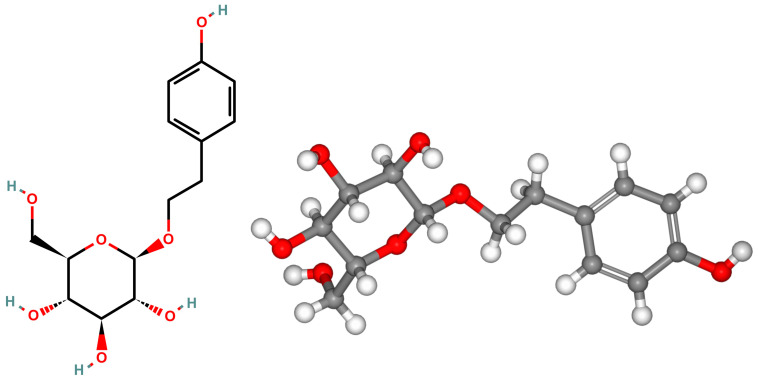
The graphical representation of salidroside (C_14_H_20_O_7_) structure. The 2D and 3D structures were accustomed according to the PubChem database (PubChem CID: 159278).

**Figure 2 nutrients-16-02387-f002:**
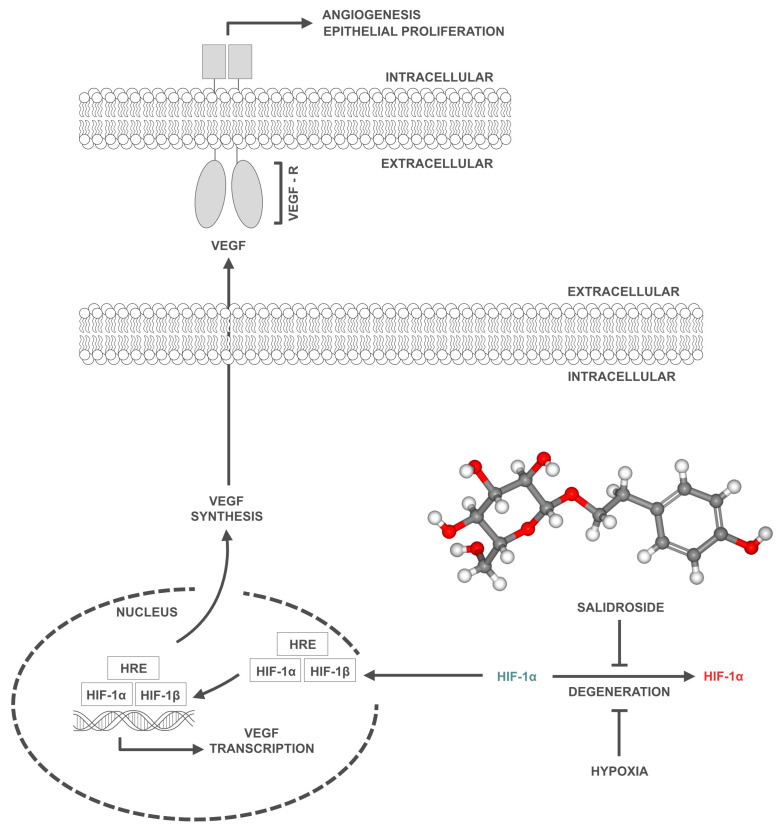
A schematic illustration of the impact of salidroside (SAL) and hypoxia on the hypoxia-inducible factor 1-alpha (HIF-1α)/vascular endothelial growth factor (VEGF) signaling pathway. SAL and hypoxia inhibit the degradation of HIF-1α (red designation, right side), leading to the accumulation of HIF-1α (green designation, left side) in the cytoplasm. Subsequently, HIF-1α translocates to the nucleus, where it dimerizes with hypoxia-inducible factor 1-beta (HIF-1β) and binds to the hypoxia-responsive element (HRE). This interaction initiates the transcription of the VEGF gene, which in turn stimulates the synthesis of VEGF in the cytoplasm. The newly synthesized VEGF is then secreted out of the cell, where it binds to VEGF receptors (VEGF-R) on the surface of endothelial cells, stimulating their proliferation and leading to the formation of new blood vessels. It is noteworthy that SAL promotes angiogenesis in a manner similar to hypoxia, but potentially offers a more favorable anabolic mechanism, as it does not induce biological stress in the cells.

**Figure 3 nutrients-16-02387-f003:**
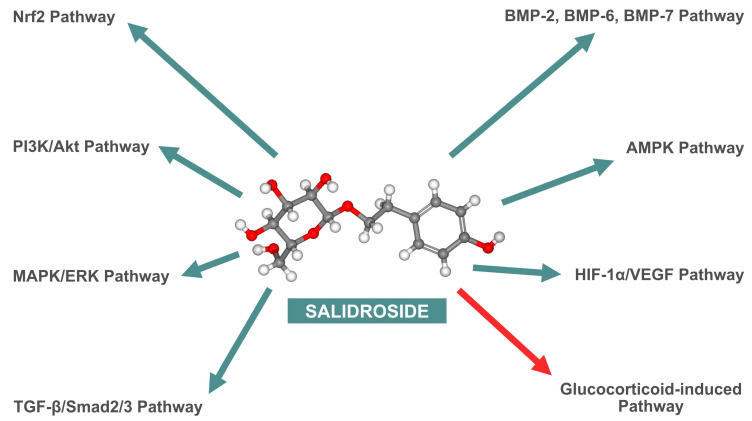
Diagram summarizing the effects of salidroside (SAL) on cellular signaling pathways. Stimulatory effects are highlighted with green arrows, whereas inhibitory effect is denoted by red arrow. Nrf2—nuclear factor erythroid 2–related factor 2; BMP-2—bone morphogenetic protein 2; BMP-6—bone morphogenetic protein 6; BMP-7—bone morphogenetic protein 7; PI3K/Akt—phosphoinositide 3-kinase/protein kinase B; AMPK—adenosine monophosphate-activated protein kinase; MAPK/ERK—mitogen-activated protein kinase/extracellular signal-regulated kinase; HIF-1α/VEGF—hypoxia-inducible factor 1-alpha/vascular endothelial growth factor; TGF-β/Smad2/3—transforming growth factor-beta/Smad family member 2/3.

**Table 1 nutrients-16-02387-t001:** Effects of SAL on various cell lines in major in vitro studies focusing on bone metabolism. rBMSCs—rat bone marrow-derived mesenchymal stem cells; ALP—alkaline phosphatase; COL1A1—collagen type I alpha 1; MG-63—human osteoblast-like cell line; ROBs—rat osteoblasts; MAPK/ERK—mitogen-activated protein kinase/extracellular signal-regulated kinase; PI3K/Akt—phosphoinositide 3-kinase/protein kinase B; C3H10T1/2—mouse pluripotent mesenchymal stem cell-like fibroblasts; MC3T3-E1—a mouse clonal; osteoblast-like cell line; BMP—bone morphogenetic protein; Smad—family of proteins that transduce extracellular signals from transforming growth factor beta (TGF-β) ligands; Runx2—runt-related transcription factor 2; OSX—osterix; BAX—B-cell lymphoma (Bcl)-2-associated X protein; CASP3—caspase-3; CASP9—caspase-9; EA.hy926—human endothelial cell line; HUVECs—human umbilical vein endothelial cells; HIF-1α—hypoxia-inducible factor 1-alpha; VEGF—vascular endothelial growth factor; AMPK—adenosine monophosphate-activated protein kinase; OCN—osteocalcin; Nrf2—nuclear factor erythroid 2-related factor 2; Keap1—Kelch-like ECH-associated protein 1; TGF-β—transforming growth factor-beta.

First Author, Year (Reference)	Cell Lines	Biological Manifestation
Pan et al., 2013 [24]	rBMSCs	Promotion of osteoblast differentiation and bone formation, upregulation of osteogenic markers ALP and COL1A1.
Guo et al., 2017 [35]	MG-63, ROBs	Enhanced proliferation and differentiation of osteoblasts, activation of MAPK/ERK and PI3K/Akt pathways, increased Runx2 and OSX expression.
Li et al., 2018 [23]	rBMSCs	Increased osteogenic differentiation, enhanced ALP activity, and mineralization, promotion of bone formation, increased expression of Runx2 and ALP.
Xue et al., 2018 [36]	ROBs	Protection against apoptosis, activation of PI3K/Akt pathway, reduction in apoptotic markers such as BAX, CASP3, and CASP9, increased expression of Runx2 and OSX.
Chen et al., 2019 [21]	C3H10T1/2, MC3T3-E1, rBMSCs	Increased proliferation and differentiation of osteoblasts, enhanced ALP activity, and mineralization via BMP/Smad pathway activation, increased expression of Runx2 and OSX.
Guo et al., 2020 [34]	EA.hy926, HUVECs, MG-63, ROBs	Increased proliferation, migration, capillary formation, and mineralization through HIF-1α/VEGF pathway activation, increased expression of ALP, Runx2, OSX, and VEGF.
Fu et al., 2022 [28]	MC3T3-E1	Enhanced proliferation and differentiation of osteoblasts, AMPK activation, inhibition of bone resorption, increased expression of ALP, COL1A1, OCN, and Runx2.
Wang et al., 2022 [37]	ROBs	Protection against oxidative stress, increased Nrf2 activation, decreased Keap1 expression, promotion of osteogenesis, increased expression of ALP and Runx2.
Xie et al., 2023 [25]	MC3T3-E1	Protection against dexamethasone-induced inhibition, increased ALP activity, activation of TGF-β/Smad2/3 pathway, increased expression of OSX and ALP.

**Table 2 nutrients-16-02387-t002:** Effects of SAL on various animal models in major in vivo studies focusing on bone metabolism. OVX—ovariectomy; RANKL—receptor activator of nuclear factor kappa-Β ligand; OPG—osteoprotegerin; HIF-1α—hypoxia-inducible factor 1-alpha; VEGF—vascular endothelial growth factor; OP—osteoporosis; SANFH—steroid-induced avascular necrosis of the femoral head; PI3K/Akt—phosphoinositide 3-kinase/protein kinase B; Bcl-2—B-cell lymphoma 2; BAX—Bcl-2-associated X protein; CASP3—caspase-3; MDA—malondialdehyde; ROS—reactive oxygen species; SOD—superoxide dismutase; GSH-Px—glutathione peroxidase; CD31—Cluster of Differentiation 31 (platelet endothelial cell adhesion molecule); KOA—knee osteoarthritis; AMPK—adenosine monophosphate-activated protein kinase; TNF-α—tumor necrosis factor alpha; IL-1β—interleukin 1 beta; IL-6—interleukin 6; Nrf2—nuclear factor erythroid 2-related factor 2; Runx2—runt-related transcription factor 2; ALP—alkaline phosphatase; OCN—osteocalcin; TGF-β—transforming growth factor-beta; Smad—family of proteins that transduce extracellular signals from transforming growth factor beta (TGF-β) ligands; COL1A1—collagen type I alpha 1.

First Author, Year (Reference)	Animal Model	Biological Effect
Pan et al., 2013 [24]	Sprague Dawley rats (OVX-induced bone loss model)	Reduced oxidative stress, increased bone mass, reduced RANKL, increased OPG.
Guo et al., 2017 [35]	BALB/c mice (tibia fracture model)	Accelerated fracture healing, enhanced osteoblast proliferation and differentiation, increased HIF-1α and VEGF.
Li et al., 2018 [23]	Sprague Dawley rats (OVX-induced OP model)	Increased bone mass and mineral apposition rates, improved bone microarchitecture, increased HIF-1α and VEGF.
Xue et al., 2018 [36]	Sprague Dawley rats (SANFH model)	Reduced osteoblast apoptosis, increased osteogenic differentiation via PI3K/Akt, increased Bcl-2, decreased BAX and CASP3.
Zheng et al., 2018 [40]	Sprague Dawley rats (OVX-induced OP model with diabetes)	Improved bone histomorphology, prevention of bone loss, upregulation of the OPG/RANKL ratio
Chen et al., 2019 [21]	C57BL/6 mice (oxidative stress model)	Preserved bone microstructure, reduced MDA and ROS, increased osteoblast function and antioxidant enzymes such as SOD and GSH-Px.
Guo et al., 2020 [34]	C57BL/6 mice (angiogenesis model using mouse embryonic metatarsals)	Enhanced endothelial sprouting, increased VEGF, increased CD31-positive endothelial cells.
Fu et al., 2022 [28]	C57BL/6 mice (KOA model)	Reduced inflammation, increased osteogenic protein expression, enhanced AMPK activation, reduced TNF-α, IL-1β, and IL-6.
Wang et al., 2022 [37]	Sprague Dawley rats (OVX-induced OP model)	Reduced oxidative stress, increased osteogenesis via Nrf2, increased Runx2, ALP, and OCN.
Xie et al., 2023 [25]	C57BL/6 mice (dexamethasone-induced OP model)	Mitigated inhibitory effects of dexamethasone on osteogenesis, activated TGF-β/Smad2/3, increased ALP and COL1A1.

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
