# Peer review of "Salidroside: A Promising Agent in Bone Metabolism Modulation"

_nutrients, 2024, doi:10.3390/nu16152387_

Round 1

Reviewer 1 Report

Comments and Suggestions for Authors

The review titled "Salidroside as a Modulator of Bone Metabolism: Insight into the Active Constituent of Rhodiola rosea" is a very good review article. Besides a few minor remarks, I believe it is definitely suitable for acceptance and publication in such a prestigious journal as a review paper. It will contribute significantly to the field. Below are my minor comments:

The title needs improvement. For a review paper, it should more clearly indicate its focus.

I am not sure if the last sentence in the abstract is necessary.

Moreover, the abstract seems to need improvement, especially its last three sentences.

"Bone loss;" - is this keyword really necessary here?

Figure 1 - the 3D model seems blurry, please correct this.

Figure 2 - the molecular structure (the gray one) is also not fully visible - please correct this.

"This implies that even at lower 247" - I don't quite understand why; I think many readers might also get confused here. Please explain this more clearly.

"The length of the capillary tubes in the presence of SAL was 46% greater than in the CM-only group" - why does this happen? In a review, I would like this to be explained.

"This protective effect was confirmed through TUNEL staining, which detects apoptotic cells, and Western blot analyses, which showed decreased levels of apoptotic markers such as Bcl-2 Associ...." - I am not sure if I can agree with this - please explain this to me.

"Nuclear factor of activated T cells, cytoplasmic 1 (NFATc1) expression related to osteoclast activity..." - This sentence should be rewritten, as it sounds somewhat awkward.

Figure 3 - poor quality of the figure. Please improve it.

The summary is very weak. It definitely needs to be entirely rewritten. Please write a proper summary, not such a general one.

I am not a native speaker, but this paper reads somewhat awkwardly, please check if the English is okay.

Overall, it's a good review but still requires a bit of work before the publication process. However, these are not major changes.

Reviewer 2 Report

Comments and Suggestions for Authors

The manuscript by Piotr Wojdasiewicz and colleagues present a potentially interesting narative review on the effect of SAL as modulator of bone metabolism. However, the manuscript suffers of major issues that should be fixed before publication.

Experimental details should be avoided (assays, reagents, etc) unless instrmental to the discussion of key findings and conclusions.

The different effects of SAL in bone cell/animal models should be discussed in a more cohesive and coordinated manner : the effect of SAL on BMP is reported in two different paragraph, paragraph 3.1.1 is about differentiation and later the effect of the compounds on other well-known differetiation markers (ALP and mineralization)

The parts on the effects of SAL on vascular endothelial cell should be better intagrated and its final effect on bone tissue better discussed

The effects of the other Rhodiola rosea secondary metabolites should be also reported and discuss as well as the effects of the total extract, if available.

Round 2

Reviewer 1 Report

Comments and Suggestions for Authors

I believe that the authors have made very good revisions to the manuscript, and it can now be accepted.

Author Response

Once again, thank you very much for taking the time to review our manuscript and providing important comments to improve it.

Reviewer 2 Report

Comments and Suggestions for Authors

 Authors only partially addressed the points raised by this reviewer:

-Experimental details should be avoided (assays, reagents, etc) unless instrumental to the discussion of key findings and conclusions. Eg. Lines 110-112 The reported experimental setup is quite basic and common for cell biologists; Lines 161-162 Is one of the available and alternative assays to evaluate cell proliferation; Lines 205-206 MTS assay is just an alternative to MTT or CCK-8. Moreover, 

In the first part of the manuscript, the authors appear to have approached the review by examining each piece of work separately. They meticulously documented the key findings of each study, but their approach lacks a cohesive, integrated analysis that synthesises the results into a unified perspective. This method may lead to a fragmented understanding of the subject, as the results are presented in isolation rather than in a contextualised framework that highlights interconnections and overarching themes. Consequently, the review may miss out on identifying broader insights and patterns that could emerge from a more synchronised and comprehensive analysis. Eg. most of the paragraphs report the details of one single study. Lines 304-310 should be moved and discussed in paragraph 3.1.1; lines 362-368 should be moved to in vitro studies part. 

-Lines 207-213. The sentences: cells exposed to CM and varying concentrations of SAL and CM supplemented with SAL are misleading. In the original paper, 3 CM were obtained by treating MG63 cells w/o SAL o with SAL at 2 concentrations, and then used to treat endothelial cells. So the effect on endothelial cells proliferation may not be a direct effect of SAL on these endothelial cells, but the effect of SAL on the cells used for the production of the CMs.
